# Doctor Clerk Implementation in Rural Community Hospitals for Effective Task Shifting of Doctors: A Grounded Theory Approach

**DOI:** 10.3390/ijerph19169944

**Published:** 2022-08-12

**Authors:** Ryuichi Ohta, Miyuki Yawata, Chiaki Sano

**Affiliations:** 1Community Care, Unnan City Hospital, 96-1 Iida, Daito-cho, Unnan 699-1221, Japan; 2Department of Community Medicine Management, Faculty of Medicine, Shimane University, 89-1 Enya-cho, Izumo 693-8501, Japan

**Keywords:** doctor clerk, task shifting, rural hospital, family medicine, primary care physician

## Abstract

With the diversification of medical care and work reform, doctor clerks play a major role today and are recruited to mitigate the burden of doctors worldwide. Their recruitment can improve the working conditions of physicians, facilitate task shifting in rural community hospitals, improve patient care, and help address the lack of healthcare resources. This study used a qualitative method to investigate difficulties in the implementation of doctor clerks and ascertain the features of effective implementation by collecting ethnographic data through field notes and semi-structured interviews with workers. We observed and interviewed 4 doctor clerks, 10 physicians, 14 nurses, 2 pharmacists, 1 nutritionist, and 2 therapists for our study. We clarified the doctor clerk process in rural hospitals through four themes: initial challenge, balance between education and expansion, vision for work progression, and drive for quality of care. We further clarified effectiveness, difficulties, and enhancing factors in implementation. Doctor clerk recruitment and bridging of discrepancies among medical professionals can mitigate professional workloads and improve staff motivation, leading to better interprofessional collaboration and patient care.

## 1. Introduction

The diversification of medical care is progressing, and task shifting among doctors is advancing worldwide, which helps reduce their burden [1,2]. Doctors’ burnout due to long working hours is also increasing, and the reform of doctors’ working styles and conditions is being discussed as an emergent issue in medical care [3,4,5]. One solution is task shifting, in which a part of the doctor’s task is transferred to other professionals [6,7]. By transferring documentation of medical practice and work carried out based on certain algorithms to other occupations, doctors’ burdens could be greatly reduced [8,9,10]. Advancing task shifting—for example, reducing clerical work and work that could be performed by other medical staff—may aid in reducing the burden on doctors and improving the quality of medical care and the health condition of patients by allowing doctors to focus more intensively on patient care.

Doctor clerks play a major role in shifting doctors’ tasks. Medical assistants have been recruited to mitigate doctors’ burden, which has led to reduced stress and improved satisfaction in working in hospitals and clinics among doctors [11,12,13]. In 2007, the Ministry of Health, Labor, and Welfare in Japan introduced the promotion of a division of roles between doctors and other medical and clerical staff [14]. In 2008, the job of doctor clerk, as one of these medical assistants, was officially established in Japan to decrease the burden on doctors. Doctor clerks could do clerical work, such as document preparation and other jobs that doctors were responsible for but could assign to others [14]. Doctor clerks were being allocated to reduce the burden on working doctors and create an environment where they could concentrate on medical treatment [15]. The work of doctor clerks varied in each medical facility, depending on the doctor’s work situation [14,15]. Since then, Japanese community hospitals have hired doctor clerks to promote task shifting for doctors. Most clerical work can now be done by doctor clerks. They can also input medical information into medical records based on doctors’ decisions. As task shifting in Japan progresses, doctor clerk competencies are expanding [14,15]. In addition to document preparation, it is possible for them to order medical tests and update electric medical records, create discharge summaries of patients, perform clerical work that contributes to improving the quality of medical care, and perform administrative work [14,15].

Effective working conditions of physicians with task shifting in rural community hospitals may improve patient care by compensating for the lack of healthcare resources. In rural Japanese community hospitals, the issues of the aging population and the lack of doctors are critical [16,17]. Implementation of doctor clerks in rural community hospitals could lead to the effective use of medical facilities. The promotion of appropriate preventive activities for doctors with the support of doctor clerks may contribute to effective health promotion in communities [18,19]. At present, there is no academic description of the methodology and difficulties of the implementation process in rural community hospitals. In the future, it will be important to promote the introduction of doctor clerks into rural community hospitals. It is, therefore, necessary to describe the introduction process, difficulties in expanding the scope of work, and methods for improving the implementation processes. The work system of doctor clerks is diverse and dependent on medical conditions [14,15]. By using qualitative research methodologies, it is possible to describe this complex work environment, its introduction process, and its effects [20]. This study clarifies the difficulties of doctor clerks in rural community hospitals and presents methods for improving the implementation process.

## 2. Materials and Methods

Our qualitative study with an action research method in a rural community hospital investigates the process of implementation of doctor clerks, its difficulties, and effective implementation requirements through ethnographic data collection with field notes and semi-structured interviews with workers.

### 2.1. Setting

Unnan city is one of the most rural cities in Japan, located in the southeast region of Shimane Prefecture. In 2020, its total population was 37,638 (18,145 men and 19,492 women), with 39% aged over 65 years, expected to reach 50% by 2025. There are 16 clinics, 12 home care stations, three visiting nurse stations, and only one public hospital (Unnan City Hospital) [21]. At the time of this study, Unnan City Hospital had 281 beds comprising 160 acute care beds, 43 comprehensive care beds, 30 rehabilitation beds, and 48 chronic care beds. There were 14 medical specialties, and the nurse-to-patient ratio was 1:10 in acute care, 1:13 in comprehensive care, 1:15 in rehabilitation, and 1:25 in chronic care.

### 2.2. Participants

Workers at Unnan City Hospital from 1 April 2016 to March 2021 were included in this study. To select participants, we employed a purposive sampling technique. Participants comprised doctor clerks, medical doctors, nurses, pharmacists, therapists, and nutritionists. We chose the participants from among the workers by considering the balance of their professional and management roles in their jobs in relation to the implementation of doctor clerks.

### 2.3. Process of Implementation of Doctor Clerks in Unnan City Hospital

Unnan City Hospital processed the implementation of two clerks for the mitigation of doctors’ office work in 2010. From then on, the doctor clerk focused on maintaining office records, such as patients’ health insurance documents, medical certificates, and long-term care insurance information.

In 2016, when the first researcher started work at the hospital, the doctor clerks’ working range expanded. Since 2019, they have started to check patients’ medical records and pick up cases lacking health maintenance, such as evidence-based cancer screening, osteoporosis, and complications of diabetes, as well as pulmonary function tests and chest X-ray among smokers. They reviewed each doctor’s documentation, test results, and lists of unperformed tests for each patient on the electric medical records to help doctors perform these tests during the patient’s next visit.

Since 2020, to reduce doctors’ documentation burdens, the clerks have taken on the responsibility of documenting patients’ discharge summaries. In Japan, these discharge summaries are mandatory for health insurance systems. The lack of a discharge summary can lead the hospital to be penalized by the government. Documentation of summaries by doctor clerks could reduce the fatigue of doctors in the hospital.

Furthermore, in 2021, as a replacement for nurses’ role doing office work for doctors, doctor clerks were allocated to internal medicine outpatients’ departments. Doctor clerks documented medical records, made patient reservations, and ordered medical tests during the next reservation of the patient’s visit on behalf of the doctors. The reallocation of nurses’ office work could reduce the burden on both physicians and nurses.

### 2.4. Measurements

#### Ethnography with Fields Notes and Semi-Structured Interviews

The first author, a specialist in family medicine and public health, performed ethnography with field notes and conducted semi-structured interviews with the participants. The researcher worked in all hospital wards and observed the interaction between doctor clerks and other workers. During the interaction between interviewees and doctor clerks regarding information sharing about patient care and physicians’ work, the first author also recorded field notes. These notes were based on the real interaction between the interviewees and doctor clerks, as well as the electronic medical records that they wrote. The first author typed the content of the field notes on a desktop computer to record them. Direct observation was conducted from 11 a.m. to noon in the internal medicine outpatient department and from 2 to 3 p.m. in the general medicine inpatient department every Monday to avoid times when the participants may have been busy. During the observation period, the researcher interviewed workers, including doctor clerks. Additionally, the first and second authors reviewed medical records related to doctor clerks’ work every Friday from 4 to 5 p.m.

The interview guide included three questions. The first question was, “What do you think about doctor clerks working in a community hospital?” Follow-up questions focused on doctor clerks’ accomplishments in a community hospital. The second question was, “What do you think about negative issues involving doctor clerks at a community hospital?” Follow-up questions focused on the concrete negative issues regarding doctor clerks. The third question was, “What do you think are the ways doctor clerks can improve work efficiency at a community hospital?” Follow-up questions focused on how doctor clerks could work effectively in a community hospital. Each interview lasted approximately 30 min, being recorded and transcribed verbatim. The transcript was shared with the interviewees to confirm the credibility of the content.

Regarding reflexivity, the first author was a family medicine doctor at Unnan City Hospital and in charge of the implementation of doctor clerks in the hospital. The second author was a doctor clerk at Unnan City Hospital who had worked as a doctor clerk from the start of the implementation. The two authors discussed the implementation of doctor clerks in community hospitals continuously from the perspectives of both doctors and doctor clerks. The first author continuously participated in discussions with various medical professionals regarding hospital administration and engaged in dialogue with them for improved administration based on mutual understanding, which could mitigate their expression of negative attitudes toward the implementation of doctor clerks.

### 2.5. Analysis

We used an inductive grounded theory approach for our analyses to build a theory of the process of doctor clerk implementation in community hospitals [22]. The analysis was performed by the first and second authors, who coded their content through discussion (initial coding) after reading the field notes and in-depth interviews. The field notes were referred to for understanding interactions between interviewees and doctor clerks in professional working conditions, especially regarding sharing information and task shifting. Codebooks were developed by the first author based on repeated readings of the research materials with respect to the initial coding for reliability. This study used process and concept coding; concepts and themes were induced, merged, deleted, and refined between the research materials and initial coding for axial coding through discussion among the first and second authors [22]. Through this discussion, axial coding was extrapolated to concepts and themes while refining the codes and themes that were created in the process of open coding. For triangulation, concepts and themes were discussed among the researchers, and in-depth interviews were analyzed iteratively during the research period after a tentative analysis of the interviews was completed for theoretical saturation. The tentative analysis results were also shared with the interviewees for an audit trial. Finally, the researchers discussed the theory, ultimately reaching an agreement on the themes. The analysis was performed in Japanese. The quotes, concepts, and themes were transcribed in English by the first and third authors, who had experience in publishing scientific qualitative studies in English and had obtained master’s degrees in English.

### 2.6. Ethical Consideration

The hospital was assured of the anonymity and confidentiality of the patients’ information. Information about this study was posted on the hospital website without disclosing any details concerning the patients. The contact information of the hospital representative was also listed on the website to address any questions regarding this study. Participants were explained the purpose of this study, and informed consent was obtained. The Clinical Ethics Committee of Unnan City Hospital has approved this study (approval code: 20220011).

## 3. Results

We observed and interviewed 4 doctor clerks (all women, average age 31 years), 10 physicians (eight men and two women, average age 36.3 years), 14 nurses (all women, average age 51.9 years), 2 pharmacists (one man and one woman, average age 49 years), 1 nutritionist (woman, age 48 years), and 2 therapists (one man and one woman, average age 49 years). Of these, one doctor clerk, three doctors, and five nurses were involved in hospital management. Through the grounded theory approach, we examined the doctor clerk implementation process in rural hospitals, consisting of 4 themes and 10 emergent concepts (Table 1). Doctor clerks faced several initial difficulties regarding their work characteristics. Other medical professionals could not understand their job profiles. Vague definitions of their job profile by medical professionals inhibited doctor clerks from working effectively. Dialogue between medical professionals mitigated their work difficulties but created an imbalance between education and work expansion. Increased workload also caused educational difficulties. Fewer doctor clerks meant sustaining the quality of work was more difficult and slowed down work expansion speed. To drive better working conditions for doctor clerks, the hospitals’ vision was essential. Hospitals’ motivation for using doctor clerks and the doctor-driven progression of the application of doctor clerks enabled the doctor clerks to work effectively. The progressive employment of doctor clerks in rural hospitals bridged discrepancies among professions, mitigated workloads of multiple professions, and improved staff motivation in rural hospitals.

### 3.1. Initial Challenge

Doctor clerk applicants faced several initial difficulties regarding their work characteristics. Other medical professionals could not understand their work content. A vague definition of their jobs amidst the existing work culture of other medical professionals inhibited the effectiveness of doctor clerks at work.

#### 3.1.1. Vague Job Description

Doctor clerks in this study were completely new to rural hospital settings. Other medical professionals did not understand their work or how to effectively administer them in the hospital context. As one of the medical doctors stated:

“The doctor clerk was effective in medical fields to reduce the burden of physicians. However, the concrete working was unknown in our hospital; so initially, we could not understand their effectiveness in our clinical situations.”

While the implementation of doctor clerks in the hospital began promptly, following the lead of other hospitals, there was no clear introduction to them of their work contents and how they would work in rural hospital contexts. Various medical professionals were confused. One of the nurses stated:

“They may not have a medical license, but they could order tests after obtaining approval from medical doctors based on the approval of medical doctors. I was confused at first in the collaboration with doctor clerks. We should have understood the systems of doctor clerks and how they could work in medical situations on behalf of medical doctors.”

The job description of a doctor clerk in a rural hospital was unclear—confusing to them and other medical professionals. Others were hesitant to ask doctor clerks to do certain jobs. One of the medical doctors stated:

“I did not know about the doctor clerk. So, I did not have any idea regarding what they can do in the medical field. I could not utilize their services properly at the initial stage.”

One of the therapists stated:

“Doctor clerks can order various prescriptions based on doctors’ orders. They can order nutritional and therapeutic therapies. This could be beneficial to speed up the treatment of patients, but I and the other physicians did not know about the ordering and effectiveness. Furthermore, while their potential could be great, the range of their assigned work was vague. So, in-depth explanation about doctor clerks should have been provided.”

The unclear work description and job responsibilities of doctor clerks impinged on the expansion of their work in rural hospitals. The official implementation of doctor clerks should be provided to more medical professionals in the hospital.

#### 3.1.2. Existing Concepts of Working

Existing working styles and contents were difficult to change immediately after the implementation of doctor clerks in the rural hospital. Many medical staff members at the rural hospital who had worked there for a long time could not change their working styles quickly. A doctor stated:

“I did not change my work style because I was used to working this way. I made an effort to write various medical records and summaries. Initially, I did not reassign such work soon.”

Work culture and customs affected doctor clerks’ work in the inpatient and outpatient departments. Doctor clerks struggled to take up the clerical work of nurses and physicians because the latter did not change their customs, such as writing medical documentations and other clerical works. One of the clerks stated:

“I can do various kinds of clerical work for medical professionals, making them more productive, and free to do other creative and meaningful work for the patients and hospital. However, they could not reassign their work soon because they were used to existing customs and did not change the work culture immediately. Some physicians persisted in their own styles of writing medical documentation. Nurses used to do some clerical work of translating vital signs and patients’ symptoms from paper to personal computers.”

One of the nurses stated:

“I could not imagine changing my working style because I was anxious about their work quality. Initially, I was very confused because they tried to do the clerical work of nurses.”

### 3.2. Balance between Education and Expansion

Dialogue among medical professionals mitigated their working difficulties, but there was an imbalance between education and expansion of their work. The increase in their work content and quantity caused difficulties in doctor clerk education. Few high-quality doctor clerks made maintaining work quality difficult and slowed down doctor clerks’ work.

#### 3.2.1. Difficulty in Clerks’ Education

The need for doctor clerks required their effective education. Doctors hoped clerks could increase their clerical jobs to enable the revision of their work styles. Additionally, hospital administrators tried to increase the number of doctor clerks for the improvement of doctors’ working conditions. However, such an increase was not easy. A doctor clerk stated:

“The work of doctor clerks varies based on the need of doctors. Without adequate education the newcomer could not work effectively to support the doctors’ work.”

Another doctor clerk stated:

“I was not used to the education of doctor clerks because I learnt the work from colleagues after starting here. I did not have any idea that I had to teach the work of a doctor clerk to others.”

Doctor clerks were used to working but unused to being educators of other doctor clerks. They struggled with establishing the doctor clerk educational system, which impinged on their usual working as doctor clerks.

#### 3.2.2. Sustaining Work Quality

Doctor clerks hoped to sustain their quality of work. However, an increase in their workload inhibited their work quality, worsening their mental condition. A doctor clerk stated:

“I am happy to know about various requirements from the doctors in this hospital. I wanted to do my best. However, the initial stage was hectic for me because the amount of work increased drastically. I could not manage all the work and was a little depressed.”

To maintain their work quality, they had to reduce the number of jobs they accepted, contrary to their will. A doctor clerk stated:

“Controlling the acceptance of job requests was inevitable. I did not want to stop accepting to maintain work quality, although it was disappointing.”

Work quality among the doctor clerks was the most valuable issue for improving healthcare in the rural hospital. Doctor clerks were, therefore, in a dilemma between their work quality and desire to accept more jobs.

#### 3.2.3. Speed of Work Expansion

The expansion of their range of work was inhibited by the creation of educational systems for doctor clerks. Work expansion speed had to be reduced to maintain a balance between the education of doctor clerks and the quality of their work. A doctor clerk stated:

“Doctor clerks’ needs for expansion were strong and urged by the administrators of the hospital; but, unfortunately, we could not meet their needs. I thought that we should satisfy their needs, but the expansion could not match our abilities and might lead to hectic situations. So, I controlled the amount and rate of accepting work from doctors.”

The effective implementation of doctor clerks required an appropriate speed of work expansion. The tendency of rapid expansion in the implementation of doctor clerks tormented them by putting them in a dilemma between their clerical and educational roles.

### 3.3. Vision for Work Progression

To drive the working conditions of doctor clerks, hospitals’ vision was essential. Hospitals’ motivation for the usage of doctor clerks and doctor driven progression of their work motivated the doctor clerks effectively.

#### 3.3.1. Overviewing Work Possibility

The effective implementation of doctor clerks in rural hospitals needs an overview of their work. Doctor clerks have a high possibility of mediating doctor workload, but quick implementation and expansion affect their work negatively. Implementation of doctor clerks in rural hospitals needed the overviewing of the possibility of working and planned expansion of their work. A doctor clerk stated:

“For the effective implementation of doctor clerks in rural hospitals, planning is essential. This time, we have experienced difficulties of dealing with the surging need for doctor clerks. To prevent the inhibition of motivation of doctor clerks, their implementation and work should be managed based on a previously established plan.”

Another doctor stated:

“The work of doctor clerks is useful for the mitigation of doctor workload. However, controlling the clerk’s work is needed because their number is limited, and their education takes time. Systematic implementation of doctor clerks to hospital could be ideal, because doctors hope to improve the quality of their work.”

A planned and systematic implementation of doctor clerks was suggested and ideal for the effective implementation of doctor clerks and doctors. Educational systems of doctor clerks needed to be planned and created before the implementation of doctor clerks.

#### 3.3.2. Doctor-Driven Progression

The working styles of doctor clerks were modified by the doctors working with them. Their fundamental roles were those of supporting doctors by mitigating their clerical work burden. Doctors’ understanding regarding doctor clerks was essential for their effective work. One doctor clerk stated:

“Doctor clerk work should be facilitated by doctors. The implementation of the system of doctor clerks can facilitate physicians’ efficient working. When they can understand the work content of doctor clerks, they can facilitate them to work effectively in their working environments.”

A doctor stated:

“By understanding doctor clerk work, my working environment got better. Through discussion with doctor clerks, I understood their roles, effectiveness, and difficulties in medical fields. The effective working of doctor clerks should be driven by doctors through discussion.”

Discussion among doctors and doctor clerks drove the collaboration between them and improved doctors’ understanding of doctor clerks. Doctor-driven building of working conditions was effective for doctor clerk work based on shared mutual understanding.

### 3.4. Driving the Quality of Care

The progressive working of doctor clerks in rural hospitals bridged the discrepancy among professions, mitigated the working load for multiple professionals, and improved the motivation of staff in rural hospitals.

#### 3.4.1. Bridging Discrepancies among Professions

The doctor clerk’s work not only reduced the burden on the doctor but also identified a gap among medical professionals. Existing medical practitioners did not notice which roles should be performed by professionals. By recruiting doctor clerks and discussing their professional roles, the attention of medical professionals was drawn to their original professional roles. A medical doctor stated:

“The doctor’s role should focus on the improvement of patient care by improving medical knowledge and skills. Although paperwork should be respected to further various clerical jobs, recruiting doctor clerks for clerical work allows doctors to focus on improving the quality of medicine. In the process, the doctor clerks can collect information about the perceptions of other professionals regarding patient care. This information could improve their collaboration in caregiving.”

The presence of doctor clerks mitigated the gap between doctors and other professionals by providing doctors with ideas from other professionals. The doctor clerks’ sharing of patients’ information with various professionals mitigated the professional hierarchy.

One of the nurses stated:

“Initially, the nurses could not accept the doctor clerk working in outpatient department because some of them might not have belief in their abilities. However, their effective work made us notice that we had to make efforts to improve our nursing skills as a medical professional.”

Another nurse stated:

“The doctor clerks communicated with the nurses. I could tell them about my perceptions and ideas of patient care. They could transmit these perceptions and ideas to the doctors. So, the collaboration among us could improve.”

The implementation of doctor clerks stimulated the awareness among medical professionals of their own professional roles and the improvements in collaboration. Effective working of doctor clerks gave various medical professionals the time to reflect on their professional roles. Doctor clerks became drivers of the quality of patient care and improvement of interprofessional collaboration.

#### 3.4.2. Workload Mitigation for Multiple Professions

The main role of doctor clerks was that of supporting doctors by taking on their clerical work. The doctor clerk’s work supported other professional work. Doctor clerks could perform clerical work more efficiently than doctors, enabling other medical professionals to understand patients’ information reports and summaries, which drove smooth and effective interprofessional collaboration. One of the nurses stated:

“A doctor clerk’s documents are easy to understand for other medical professionals. The doctors’ descriptions may be difficult for other professionals. The doctor clerk could become a kind of translator of medical information for other medical professionals. This information transfer is improving interprofessional collaboration in the hospital.”

One of the doctor clerks stated:

“I try to ask doctors when I cannot understand their writing on the medical charts. The patients’ summary and reports on medical records are essential for other medical professionals. Their preciseness and understandability are important.”

The work of doctor clerks has changed doctors’ way of recording information on patients’ medical charts. Understanding patients’ past and present conditions improves communication among medical professionals in rural hospitals, which could improve patient care.

#### 3.4.3. Improving Staff Motivation

Various medical professionals were compensated for doctor work in the rural hospital. The task shifting from doctors to doctor clerks mitigated other professionals’ burdens. This improved their professional motivation. One of the nurses stated:

“Thanks to doctor clerks, nurses are focusing on their profession such as the education of patients’ self-management and support for endoscopies. The doctor clerk’s functions should be encouraged for better patient care.”

One of the therapists stated:

“Their clerical work of ordering rehabilitation increased the number of patients who could get rehabilitation therapies smoothly. In previous situations, therapists had to tell doctors to order rehabilitation, but now, doctor clerks advance rehabilitation therapy using a doctor made template. Our burden of suggesting rehabilitation was reduced.”

Task shifting by doctor clerks improved the motivation of other medical professionals to focus on their own professionality, which could improve patient care in the rural hospital.

## 4. Discussion

This qualitative study, through the grounded theory approach, clarified the process of doctor clerk implementation in rural hospitals regarding effectiveness, difficulties, and enhancing factors. The introduction of doctor clerks initially faced poor understanding from medical professionals because of vague job definitions and preexisting conceptions of medical work. Dialogue among medical professionals mitigated their work difficulties, but there was an imbalance between education and work expansion. To improve the working conditions of doctor clerks, realizing their effectiveness and doctor-driven progression for clerk work were essential for motivating them to work effectively. The progressive working of doctor clerks in rural hospitals bridged this discrepancy among professions, mitigated the working load for multiple professionals, and improved the staff motivation in rural hospitals.

The effective implementation of doctor clerks needs the involvement of various medical professionals in the initial stages. The doctor clerk role is a relatively new one, especially in rural Japan. The understanding of various medical professionals in hospitals is mandatory for the work progress of doctor clerks. Task shifting has been performed worldwide to reduce the work burden on doctors in hospitals since the implementation of medical assistants and other support workers has been advanced [23,24]. However, prompt implementation of such jobs in clinical situations could confuse existing workers [25,26]. It is important to consider the effects of new job implementation on the existing work cultures and in contexts of medical situations. This study shows that resistance to changing working styles among medical professionals could impinge on new workers’ effective functioning. For the effective implementation of supportive workers such as doctor clerks, medical institutions and doctors should discuss their present work and ways doctor clerks can change their work styles effectively [27,28]. Future studies could investigate the effective implementation of doctor clerks and the effectiveness of their work in rural hospitals where medical professionals are lacking.

For the expansion of the role of doctor clerks, educational systems for them should be established through discussions among medical professionals to reduce physical and mental stress among doctor clerks. This study shows how the education of doctor clerks impairs the progression of doctor clerk work in the rural hospital. The educational system of doctor clerks may depend on their work contexts, and their basic abilities vary in Japanese doctor clerk systems [29]. For effective education, doctor clerk competencies should be defined in each hospital, fitting the needs of medical staff [30,31]. We show that sustaining and improving the work quality of doctor clerks was hoped for by both clerks and other medical staff. For high-quality doctor clerks, their competencies in rural hospitals should be established. The establishment of the competencies could drive their work effectively, speeding up the expansion of their work in hospitals.

Planned implementation and future vision for doctor clerks’ work should be driven by doctors and based on leadership to promote the roles and status of doctor clerks in hospitals. For the effective implementation of doctor clerk systems, an overview and blueprints of the implementation should be devised [32]. The motivation of doctor clerks and the breadth of their work can vary depending on the hospital’s plan for their implementation. The trend of task shifting from medical doctors to other medical professionals is encouraged in developed countries; hence, community hospitals must change present systems promptly [33,34]. The force of change is confusing to medical professionals in rural contexts where medical professionals are lacking [33,34]. Without leadership, task shifting can deteriorate the present situation [35]. Moreover, doctors should be motivated regarding the implementation of doctor clerks. The professional hierarchy is still present among medical professionals in rural contexts, and other medical professionals cannot change doctors’ working conditions [36,37]. In rural contexts, to effectively drive task shifting, doctors should embark on the implementation of doctor clerks. Until now, doctor-driven implementation of doctor clerks and system building might not have been reported to include the quality improvement of patient care. Subsequent studies can assess the effectiveness of the doctor-driven implementation of doctor clerks in rural contexts.

The effective implementation of doctor clerks can improve the quality of patient care by mitigating the burden of various medical professionals and focusing their attention on their professional roles. The implementation of doctor clerks could connect medical doctors and other medical professionals. The new addition of doctor clerks could posit new insights into the professional roles of medical professionals. Their presence in rural hospitals could drive professional identity formation, driving their motivation as professionals [38,39]. As present medical systems are complicated, medical professionals may be overwhelmed by clerical work, losing their identity and motivation [9,40]. This article shows that the mitigation of the burden of medical professionals could drive their motivation to work as a professional. The effective implementation of doctor clerks and task shifting could improve interprofessional collaboration and patient care. In the present medical system, the effective implementation of doctor clerks might not be encouraged by the government in rural Japan [41]. For effective task shifting, governments and hospitals should introduce effective educational and implementation systems for doctor clerks, facilitating their introduction into community hospitals lacking medical professionals. Our research results can drive the trend.

This study has certain limitations. First, there is the issue of the interviewer–participant relationship. The participants might have faced difficulties because the interviewer was a medical doctor familiar with doctor clerks who usually worked in the hospital. The researcher tried to perform interviews with multiple participants in various situations and ensure that participants did not feel conscious of their hospital reputations. Second, transferability is another limitation of this study. This research was performed in one rural hospital. To improve the reliability of the study, we used audit trials and purposive sampling and maintained a long duration for collecting data from participants across five years. The long period of observation could also have affected our results because of the changes in medical conditions, such as the retirement of medical staff and the employment of new staff. In this study, the interviewees had worked continuously for five years, which could contribute to the reliability of the study. Future studies should investigate the effects of the implementation of doctor clerks in other regions and international contexts. Third, the interview transcripts were coded by the first author, which could affect this study’s credibility. Nevertheless, to improve the quality of the current research, the second author reviewed the process of coding, concepts, and themes as theoretical triangulation.

## 5. Conclusions

This study clarified the implementation process of doctor clerks in rural hospitals: its effectiveness, difficulties, and enhancing factors. Doctor-driven implementation of doctor clerks and doctor clerks’ bridging of discrepancy of various professionals is essential to its effectiveness. Bridging discrepancies among professions can mitigate workloads for multiple professions and improve staff motivation, leading to better interprofessional collaboration and patient care.

## Figures and Tables

**Table 1 ijerph-19-09944-t001:** Results of the Grounded Theory Approach regarding the implementation of doctor clerks.

Theme	Concepts
Initial challenge	Vague job description
	Existing concepts of work
Balance between education and expansion	Difficulty in education of clerks
	Sustaining the quality of work
	Speed of expanding work
Vision for progression of work	Overview of the possibility of working
	Doctor-driven progression
Driving the quality of care	Bridging the discrepancy between professions
	Workload mitigation across multiple professions
	Improving staff motivation

## Data Availability

The datasets used and/or analyzed during the current study may be obtained from the corresponding author upon reasonable request.

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
