# Peer review of "Doctor Clerk Implementation in Rural Community Hospitals for Effective Task Shifting of Doctors: A Grounded Theory Approach"

_ijerph, 2022, doi:10.3390/ijerph19169944_

Round 1
Reviewer 1 Report
The "material and method" section is missing:
- an explanation of the concept of "axial coding", how was this coding done? - explanation of the concept of "an audit trial", who was it and what was their role in this study?
- the characteristics of the studied group are missing: age, sex, education, etc.
- an explanation of the term "grounded theory", how did the researchers cope with this theory?
In the discussion, please refer to the fact that the study lasted from 2016 to 2021 (5 years). In the discussion, the authors gave 3 years (see line 464). Does such a long period of the study have no effect on the results? Has the condition of the medical staff not changed during this time?
In addition, there is also no comment on the hospital management staff? Why were they not included in the study? Did they know about the problems of medical personnel related to the position of doctor clerks? The decisions of the management staff have a significant impact on the environment of employees.
Author Response
Responses to the reviewers’ comments
Thank you for reviewing our manuscript and providing suggestions for its improvement. We have provided point-by-point responses to the reviewers’ comments. Our revisions are indicated in red font here and in the manuscript. We hope that the revised manuscript meets the journal’s requirements and can now be considered for publication.
The "material and method" section is missing:
Response:
Thank you for your valuable feedback. Per your comment, we have added a Materials and Methods section (Line 73).
- an explanation of the concept of "axial coding", how was this coding done? - explanation of the concept of "an audit trial", who was it and what was their role in this study?
Response:
Thank you for your valuable feedback. Per your comment, we have added in-depth information about our analysis in the Analysis section, regarding axial coding, audit trial, and each author’s role (Lines 154–174).
- the characteristics of the studied group are missing: age, sex, education, etc.
Response:
Thank you for your valuable feedback. Per your comment, we have added information about the gender and age of the participants in the Results section (Lines 185–190).
- an explanation of the term "grounded theory", how did the researchers cope with this theory?
Response:
Thank you for your valuable feedback. Per your comment, we have added information about the grounded theory approach in the Methods section, regarding how to deepen concepts and themes in theory-building (Lines 154–174).
In the discussion, please refer to the fact that the study lasted from 2016 to 2021 (5 years). In the discussion, the authors gave 3 years (see line 464). Does such a long period of the study have no effect on the results? Has the condition of the medical staff not changed during this time?
Response:
Thank you for your valuable feedback. Per your comment, we have revised the Limitations portion of the Discussion section to include the negative effect of longer observation periods on the results of this study (Lines 512–518).
In addition, there is also no comment on the hospital management staff? Why were they not included in the study? Did they know about the problems of medical personnel related to the position of doctor clerks? The decisions of the management staff have a significant impact on the environment of employees.
Response:
Thank you for your valuable feedback. We agree with your suggestion. Actually, the participating doctors and nurses were members of the hospital management staff. Per your comment, we have added information about the participants in the Results section (Lines 185–190).
Reviewer 2 Report
Ohta and colleagues report on a qualitative study reporting on the implementation of doctor clerks in a rural community hospital.
Substantial language editing is required to make this readable. The are key terms used by the authors which are used incorrectly and suggest an unfamiliarity with the literature in the field of implementation science, e.g. the authors refer to the “application process” of doctor clerks, which would more correctly be termed their implementation.
In the introduction the authors refer to the use of action research methodologies, but make no further reference to how these methodologies were used. Was there any theory used underpinning this research?
Further detail is required in the methods to understand how this study was conducted – please refer to the additional comments below.
Ln 28 – being promoted by whom?
Ln 35 – not made explicit that advancing task shifting would improve the quality of medical care
Ln 36-38 - This seems counterintuitive - if number of doctors in hospitals and clinics has reduced, then how has the burden on doctors been mitigated?
Ln 56 - Aging of the population or of the doctors?
Ln 65 – use of action research methodologies – there is no subsequent reference to action research methodologies and how a framework for action research underpinned this study
Ln 84 – more detail is needed regarding how interview participants were selected – it is stated that all workers were included, and in the discussion there is reference to purposive sampling, but no description of how participants were selected; did all workers across the entirety of Unnan City Hospital between 2016 and 2021 provide informed consent? This seems implausible.
Ln 112 - Is this researcher also a health professional at this hospital? If so, how do they think their existing relationship with participants may have impacted on the data collection? What methods were used to try to mitigate any impact? While this is referred to in the discussion, these details are required in the methods section.
Ln 114 - What were the field notes relating to? More detail is required. Was a standardised tool used for data collection?
Ln 115 – More detail is required regarding the observation period. For example, how long was the observation period? How long for each hospital ward? During what time of day? It could be expected that interactions would vary at different times of the day/week.
Ln 129 – who did the coding?
Quotes in findings: Have these quotes been translated? This needs to be specified if so. Who translated them? How is it verified that the original meaning has been preserved?
Ln 189-204 – This concept does not seem sufficiently different to “Understanding work content” to warrant its own concept
Ln 216 – which specific customs are you referring to that became a barrier?
Author Response
Responses to the reviewers’ comments
Thank you for reviewing our manuscript and providing suggestions for its improvement. We have provided point-by-point responses to the reviewers’ comments. Our revisions are indicated in red font here and in the manuscript. We hope that the revised manuscript meets the journal’s requirements and can now be considered for publication.
Ohta and colleagues report on a qualitative study reporting on the implementation of doctor clerks in a rural community hospital.
Substantial language editing is required to make this readable. The are key terms used by the authors which are used incorrectly and suggest an unfamiliarity with the literature in the field of implementation science, e.g. the authors refer to the “application process” of doctor clerks, which would more correctly be termed their implementation.
Response:
Thank you for your valuable feedback. Per your comment, we have changed “application” to “implementation” across the manuscript. In addition, our manuscript has been reviewed and revised by an English-language academic editing company.
In the introduction the authors refer to the use of action research methodologies, but make no further reference to how these methodologies were used. Was there any theory used underpinning this research?
Response:
Thank you for your valuable feedback. We agree with your comment. The current study involved qualitative research for theory-building, not concrete action research. Per the comment, we have revised the description in the Introduction section to refer to qualitative research rather than action research (Lines 69–70).
Further detail is required in the methods to understand how this study was conducted – please refer to the additional comments below.
Ln 28 – being promoted by whom?
Response:
Thank you for your valuable feedback. Per your comment, we have revised the phrase for better understanding as follows:
“The diversification of medical care is progressing, and task shifting among doctors is advancing worldwide, which helps reduce their burden.” (Lines 28–29).
Ln 35 – not made explicit that advancing task shifting would improve the quality of medical care
Response:
Thank you for your valuable feedback. Per your comment, we have added information about concrete task-shifting and its effect on doctors’ working conditions and their improved focus on patient care, leading to better patient care (Lines 34–38).
Ln 36-38 - This seems counterintuitive - if number of doctors in hospitals and clinics has reduced, then how has the burden on doctors been mitigated?
Response:
Thank you for your valuable feedback. We apologize for this error. Per your comment, we have added information about the effect of task-shifting on doctors’ stress and satisfaction as follows:
“Doctor clerks play a major role in shifting doctors’ tasks. Medical assistants have been recruited to mitigate doctors’ burden, which has led to reduced stress and improved satisfaction of working in hospitals and clinics among doctors.”(Lines 39–41).
Ln 56 - Aging of the population or of the doctors?
Response:
Thank you for your valuable feedback. Per your comment, we have revised the sentence for better understanding as follows:
“In rural Japanese community hospitals, the issues of the aging population and the lack of doctors are critical.” (Lines 59–60).
Ln 65 – use of action research methodologies – there is no subsequent reference to action research methodologies and how a framework for action research underpinned this study
Response:
Thank you for your valuable feedback. We agree with your comment. This was a qualitative research for theory-building, not concrete action research. Per the comment, we have revised the description in the Introduction section from “action research” to “qualitative research” (Lines 69–70).
Ln 84 – more detail is needed regarding how interview participants were selected – it is stated that all workers were included, and in the discussion there is reference to purposive sampling, but no description of how participants were selected; did all workers across the entirety of Unnan City Hospital between 2016 and 2021 provide informed consent? This seems implausible.
Response:
Thank you for your valuable feedback. We agree; according to the comment, we have added information about the purposive sampling method and respecting the participants’ roles, such as professional and management roles, as follows:
“Workers at Unnan City Hospital from April 1, 2016 to March 2021 were included in this study. To select participants, we employed a purposive sampling technique. Participants comprised doctor clerks, medical doctors, nurses, pharmacists, therapists, and nutritionists. We chose the participants from among the workers by considering the balance of their professional and management roles in their jobs, in relation to the implementation of doctor clerks.” (Lines 89–94).
Ln 112 - Is this researcher also a health professional at this hospital? If so, how do they think their existing relationship with participants may have impacted on the data collection? What methods were used to try to mitigate any impact? While this is referred to in the discussion, these details are required in the methods section.
Response:
Thank you for your valuable feedback. Per your comment, we have added a description of the authors’ reflexivity and how their expression of negative attitudes toward implementation of doctor clerks can be mitigated as follows:
“Regarding reflexivity, the first author was a family medicine doctor at Unnan City Hospital and in charge of implementation of doctor clerks in the hospital. The second author was a doctor clerk at Unnan City Hospital, who had worked as a doctor clerk from the start of the implementation. The two authors discussed the implementation of doctor clerks in community hospitals continuously, from the perspectives of both doctors and doctor clerks. The first author continuously participated in discussions with various medical professionals regarding hospital administration and engaged in dialogue with them for improved administration based on mutual understanding, which could mitigate their expression of negative attitudes toward implementation of doctor clerks.”
(Lines 144–152).
Ln 114 - What were the field notes relating to? More detail is required. Was a standardised tool used for data collection?
Response:
Thank you for your valuable feedback. Per your comment, we have added information regarding the field notes and the data collection in the Methods section as follows:
“The first author, a specialist in family medicine and public health, performed ethnography with field notes and conducted semi-structured interviews with the participants. The researcher worked in all hospital wards and observed the interaction between doctor clerks and other workers. During the interaction between interviewees and doctor clerks regarding information sharing about patient care and physicians’ work, the first author also recorded field notes. These notes were based on the real interaction between the interviewees and doctor clerks, as well as the electronic medical records that they wrote. The first author typed the content of the field notes on a desktop computer to record them. Direct observation was conducted from 11 a.m. to noon in the internal medicine outpatient department and 2 to 3 p.m. in the general medicine inpatient department every Monday to avoid times when the participants may have been busy. During the observation period, the researcher interviewed workers, including doctor clerks. Additionally, the first and second authors reviewed medical records related to doctor clerks’ work every Friday from 4 to 5 p.m.” (Lines 120–133).
Ln 115 – More detail is required regarding the observation period. For example, how long was the observation period? How long for each hospital ward? During what time of day? It could be expected that interactions would vary at different times of the day/week.
Response:
Thank you for your valuable feedback. We agree with the comment. Accordingly, we have added information about the timing and duration of ethnography and observation (Lines 120–132).
Ln 129 – who did the coding?
Response:
Thank you for your valuable feedback. Per your comment, we have added detailed information about our analysis in the “Analysis” section, specifying axial coding, audit trial, and each author’s role (Lines 154–174).
Quotes in findings: Have these quotes been translated? This needs to be specified if so. Who translated them? How is it verified that the original meaning has been preserved?
Response:
Thank you for your valuable feedback. According to your comment, we have added information about the translation from Japanese to English in the analysis section. For the translated parts, we have tried to preserve the original meaning. The added description is as follows:
“The analysis was performed in Japanese. The quotes, concepts, and themes were transcribed in English by the first and third authors, who had experience in publishing scientific qualitative studies in English and had obtained master’s degrees in English.” (Lines 172–174).
Ln 189-204 – This concept does not seem sufficiently different to “Understanding work content” to warrant its own concept
Response:
Thank you for your valuable feedback. We agree with your comment; accordingly, we have merged the two concepts in the “Vague job description” section (Lines 212–243).
Ln 216 – which specific customs are you referring to that became a barrier?
Response:
Thank you for your valuable feedback. We agree with your suggestion. Accordingly, we have added the customs followed by medical professionals as a barrier to doctor clerk work (Lines 253–263).
Reviewer 3 Report
Nixe paper.
Promotion of doctor clerk will facilitate task management among medical staff.
I hope your further advance in this field and nextpapre.
Author Response
Responses to the reviewers’ comments
Thank you for reviewing our manuscript and providing suggestions for its improvement. We have provided point-by-point responses to the reviewers’ comments. Our revisions are indicated in red font here and in the manuscript. We hope that the revised manuscript meets the journal’s requirements and can now be considered for publication.
Nice paper.
Promotion of doctor clerk will facilitate task management among medical staff.
I hope your further advance in this field and next paper.
Response:
Thank you for your valuable feedback. Per your comment, we have added various information for promoting future research in the Discussion section (Lines 497–512).
Round 2
Reviewer 2 Report
Thank you to the authors for clearly addressing my comments. I have one remaining comment regarding the use of the term "action research method" - while the authors have removed this term from the main body of text, it remains in the abstract (lines 15-16), and should be amended.
Author Response
Responses to the reviewer’s comments
Thank you for reviewing our manuscript and providing suggestions for its improvement. We have provided point-by-point responses to the reviewer’s comments. Our revisions are indicated in red font here and in the manuscript. We hope the revised manuscript meets the journal’s requirements and can now be considered for publication.
Thank you to the authors for clearly addressing my comments. I have one remaining comment regarding the use of the term "action research method" - while the authors have removed this term from the main body of text, it remains in the abstract (lines 15-16), and should be amended.
Response:
Thank you for giving us a productive comment. We agree with the reviewer's suggestion and deleted the phrase action research method and added the qualitative method.